# The Role of Microbiota in the Pathogenesis of Esophageal Adenocarcinoma

**DOI:** 10.3390/biology10080697

**Published:** 2021-07-22

**Authors:** Megan R. Gillespie, Vikrant Rai, Swati Agrawal, Kalyana C. Nandipati

**Affiliations:** 1School of Medicine, Creighton University, 2500 California Plaza, Omaha, NE 68178, USA; meganrgillespie@outlook.com; 2Department of Translational Research, College of Osteopathic Medicine of the Pacific, Western University of Health Sciences, Pomona, CA 91766, USA; vrai@westernu.edu; 3Department of Surgery, Creighton University School of Medicine, 2500 California Plaza, Omaha, NE 68178, USA; swatiagrawal@creighton.edu

**Keywords:** esophageal adenocarcinoma, microbiota, molecular pathogenesis

## Abstract

**Simple Summary:**

Esophageal adenocarcinoma has a poor 5-year survival rate and is among the highest mortality cancers. Changes in the esophageal microbiome have been associated with cancer pathogenesis; however, the molecular mechanism remains obscure. This review article critically analyzes the molecular mechanisms through which microbiota may mediate the development and progression of esophageal adenocarcinoma and its precursors-gastroesophageal reflux disease and Barrett’s esophagus. It summarizes changes in esophageal microbiome composition in normal and pathologic states and subsequently discusses the role of altered microbiota in disease progression. The potential role of esophageal microbiota in protecting against the development of esophageal adenocarcinoma is also discussed. By doing so, this article highlights specific directions for future research developing microbiome-mediated therapeutics for esophageal adenocarcinoma.

**Abstract:**

Esophageal adenocarcinoma (EAC) is associated with poor overall five-year survival. The incidence of esophageal cancer is on the rise, especially in Western societies, and the pathophysiologic mechanisms by which EAC develops are of extreme interest. Several studies have proposed that the esophageal microbiome may play an important role in the pathophysiology of EAC, as well as its precursors—gastroesophageal reflux disease (GERD) and Barrett’s esophagus (BE). Gastrointestinal microbiomes altered by inflammatory states have been shown to mediate tumorigenesis directly and are now being considered as novel targets for both cancer treatment and prevention. Elucidating molecular mechanisms through which the esophageal microbiome potentiates the development of GERD, BE, and EAC will provide a foundation on which new therapeutic targets can be developed. This review summarizes current findings that elucidate the molecular mechanisms by which microbiota promote the pathogenesis of GERD, BE, and EAC, revealing potential directions for additional research on the microbiome-mediated pathophysiology of EAC.

## 1. Introduction

Esophageal cancer accounts for approximately 3.2% of cancer cases worldwide [1]. It is responsible for 1 in every 20 cancer deaths and ranks sixth among the highest mortality cancers [1]. The two predominant histological subtypes of esophageal cancer are squamous cell carcinoma (ESCC) and esophageal adenocarcinoma (EAC). Historically, ESCC accounted for a greater percentage of esophageal cancer than EAC, but recent studies anticipate increasing global EAC incidence [2]. Currently, the main histological subtype of esophageal cancer in Western societies is EAC [3]. The overall incidence of EAC increased six-fold in the United States between 1975–2001 [4]. Since 2001, EAC incidence has stabilized in the United States at an estimated 18,440 new EAC cases and 16,170 estimated deaths in 2020 alone [5]. The five-year survival rate for EAC remains poor at 21.4%, which is often attributed to the advanced stage at the diagnosis [6]. The increased incidence of EAC and low five-year survival rate have caused an increasing interest in determining the pathogenesis of EAC. 

EAC develops in the distal esophagus in approximately 75% of cases [7]. Normally, the distal esophagus is lined by non-keratinized squamous epithelium. It transitions to a simple columnar epithelium at the gastroesophageal junction where protection against stomach acids is needed. Considering EAC develops from columnar glandular cells, conditions that predispose changes from this normal squamous epithelial lining of the distal esophagus to columnar metaplasia increase the risk of EAC [8]. Conditions that promote this transition include a reflux-mediated cycle from gastroesophageal reflux disease (GERD) to Barrett’s esophagus (BE), to EAC, termed the “EAC cascade” [9]. 

The EAC cascade begins with GERD, a pathologic condition in which gastric contents reflux into the esophagus [10,11]. Anatomic factors that mediate GERD development include crural diaphragm disruption, also known as hiatal hernias, and disruption of the lower esophageal sphincter [10]. Apart from the abovementioned anatomical factors, neurological and salivary disorders that disrupt esophageal motility can also contribute to GERD pathophysiology. Specifically, saliva contains bicarbonate that aids the neutralization of refluxed stomach acids. Rat-model studies demonstrated that removing salivary glands disrupts esophageal mucosa integrity and increases esophageal mucosa permeability to damaging hydrogen ions [12,13]. Thus, a change in saliva content may affect esophageal mucosa integrity. Dysregulation of gastrointestinal enzymes and hormones may also play an important role in the development of GERD and its related complications [10,14,15,16]. Gastric emptying disorders, increased abdominal pressure, or defective esophageal peristalsis can also contribute to GERD development [17]. Each of these mechanisms can lead to refluxate entering the esophagus where the acid, pepsin, bile salts, and pancreatic enzymes contained in the refluxate damage the normal squamous epithelial lining of the distal esophagus [18]. Acid in particular has been shown to degrade the esophageal mucosa and lead to dilated intercellular spaces [19]. When chronically exposed to this acidic refluxate, the esophagus is driven into an inflammatory cycle of epithelial cell damage and regrowth [20]. Studies have reported that weekly GERD symptoms increase the odds of EAC five-fold [21]. Furthermore, the damage-proliferation cycle of GERD can predispose esophageal tissue to BE, the second step in the reflux-mediated cascade to EAC [22,23,24]. 

BE is characterized by a metaplastic transition from reflux-damaged squamous epithelium to mucus-secreting columnar cells. This metaplastic change is considered a pre-cancerous epithelium that increases the risk of EAC 30 to 125-fold [20,25]. Considering this association, GERD and BE are clear risk factors for EAC; however, GERD leads to genetic alterations in BE cells and progression to EAC at a rate of only 0.12–0.6% per year [26]. EAC can also develop independently of GERD or BE. Therapeutic targets for these risk factors have yet to significantly improve the outcomes of patients with EAC. Consequently, new molecular targets for EAC therapies are of great interest. 

Microbiota have been implicated as independent factors that contribute to GERD, BE, and EAC development. The term ‘microbiota’ refers to the commensal organisms, primarily bacteria, residing in the human gastrointestinal tract. Normally, microbiota play a mutualistic role within the human gastrointestinal tract. However, when microbiota and host immunity become imbalanced, or microbiota maladapt to the host environment, dysbiosis develops [27]. The relationship between dysbiosis and cancer is critical considering microbiota are capable of affecting tumor-promoting inflammation, genomic instability, mutations, proliferative signaling, and immune system evasion [27,28,29]. Dysbiosis is already implicated in colon, gastric, pancreatic, laryngeal, breast, and gallbladder cancers [30]. In the past two decades, research on the role of microbiota in EAC has surged due to the development of 16S rRNA and rDNA sequencing technologies. This review article will summarize the current findings from various studies that elucidate the molecular mechanisms by which microbiota promote the pathogenesis of GERD, BE, and EAC. 

## 2. Microbiota Expression: Normal vs. Pathologic

### 2.1. Mutualistic and Dysbiotic Roles of Microbiota

Microbiota play a mutualistic role by aiding the host in digestion, vitamin synthesis, immune system development, and drug metabolism in exchange for the host’s nutrients and secure environment [31]. Microbiota can also protect the host tissue from pathogenic invasion and contribute to the structural integrity of the esophageal mucosal barrier [31]. For this mutualistic relationship to occur, the host’s immune system must tolerate the microbiota, and the commensal microbiota must adapt to changes in the host’s microenvironment. If either of these is not maintained, there is a change in the composition and function of the microbiota, a phenomenon known as dysbiosis. 

Dysbiosis can arise as a result of two general mechanisms: a loss of tolerance by the host immune system or maladaptation of the microbiota to the host’s environment. A loss of tolerance by the host immune system induces dysbiosis considering the mutualistic microbiota become targets of the host innate and adaptive immune system [32]. With subsequent immune system reactions and inflammation, the composition of microbiota is altered. This leads to a loss of microbial functions that are vital to the host. Dysbiosis can also result from the inability of the microbiota to adapt to the host’s microenvironment. Variations in the host microenvironment can be induced by saliva, inflammation, infection, diet, or xenobiotics and consequently influence gastrointestinal dysbiosis [27,29,33]. Saliva can mediate the esophageal microbiome environment considering it decreases esophageal mucosa permeability to acid, restores physiologic esophagus pH through buffering, strengthens the pre-epithelial barrier of the esophageal mucosa, and contains epidermal growth factor that is essential for healing esophageal mucosal lesions [12,34,35]. Inflammation is a particularly well-established mediator of dysbiosis [33]. Inflammation mediates dysbiosis through positive selection of microbial species with characteristics advantageous for the inflamed microenvironment and negatively selects against microbiota without factors that enable them to thrive in the inflamed microenvironment. The inflammatory environment seen in the EAC cascade may initiate dysbiosis in this manner. 

Once established, dysbiotic microbiota can manipulate host immune system mechanisms, such as inflammasome and TLR signaling, in order to maintain the dysbiotic composition and function of the microbiota [27]. This manipulation of the immune system, in addition to microbial mediation of metabolism, cellular proliferation, and inflammation, makes dysbiotic microbiota key factors in the initiation and development of cancer [28]. Consequently, determining the role of microbiota in the pathogenesis of the EAC cascade is of great consequence. 

### 2.2. Microbiota Composition in Healthy Esophageal Tissue

There are an estimated 10^14^ bacteria in the human digestive tract [36]. In the distal esophagus alone, there are 10^4^ bacteria per mm^2^ of mucosal surface [37]. The density and composition of microbiota varies from the oral cavity to the colon. Studies have demonstrated that the esophageal microbiome has distinct resident microbiota that are distinct from—but similar to—the microbiome found in the oral mucosa [37,38]. The distinct composition of the esophageal microbiome community may be explained by predominantly resident, rather than transient, microbiota. Pei et al.’s observation of a close association between bacteria and the cell surface of the mucosal epithelium lining the esophagus supports this possibility [39]. 

Several studies have used culture-dependent and culture-independent analysis methods to determine the composition of the healthy esophageal microbiome (Table 1). Despite differences in sample collection and analysis methods, all studies found that *Streptococcus* is one of the dominant genera in the normal esophagus—a genus of Gram-positive, predominantly facultative anaerobes that are catalase-negative. Two studies noted *Streptococcus* as the genera with the greatest relative abundance [37,40]. Peter et al. uniquely noted the dominant genus was *Tissierella soehngenia*—a Gram-positive anaerobe genus normally associated with fecal tracts [41,42].

Several studies demonstrate that the esophageal microbiome did not express site-specific bacteria when comparing proximal, middle, and distal esophageal segments [43,44]. This consistent microbial pattern along the length of the normal esophagus was shown to have variable composition when comparing study subjects. Genetic and lifestyle factors may contribute to this esophageal microbiome variability between healthy study subjects [43]. Under pathologic conditions, this microbial pattern of the esophagus is altered.

**Table 1 biology-10-00697-t001:** Microbiota expressed by normal esophagus.

Article	Publication Year	Method	Sample Size (n)	Number of Identified Genera	Dominant Phyla	Dominant Genera
Peter et al. [42]	2020	16S rRNA gene DNA sequencing	12	N/A	*Firmicutes* (47.81%), *Proteobacteria* (20.67%), *Bacteroidetes* (16.93%), *Actinobacteria* (5.57%), *Fusobacteria* (4.76%)	*Tissierella soehngenia* (16.67%) *Lactobacillus* (7.15%) *Streptococcus* (7.27%) *Acinetobacter* (5.80%) *Prevotella* (5.24%)
Dong et al. [43]	2018	16S rRNA Illumina sequencing	27	594	*Firmicutes* (37.42%) *Proteobacteria* (43.61%) *Bacteroidetes* (13.17%) *Actinobacteria* (2.53%) *Fusobacteria* (1.22%) *TM7* (1.06%)	*Streptococcus* *Neisseria* *Prevotella* *Actinobacillus* *Veillonella*
Blackett et al. [45]	2013	Selective media cultures + 16S rRNA PCR	39	19	N/A	*Streptococcus* *Prevotella* *Staphylococcus* *Rothia* *Actinomyces* *Bifidobacterium* *Staphylococcus Neisseria*
Yang et al. [46]	2009	16S rRNA Sanger Sequencing	12	N/A	*Firmicutes*	*Streptococcus* (78.750%) *Prevotella* (4.300%) *Gemella* (3.400%) *Veilonella* (3.075%) *Pasteurellaceae* (2.075%) *Haemophilus* (1.925%) *Rothia* (1.025%)
Zilberstein et al. [40]	2007	Selective media cultures	10	N/A	N/A	*Streptococcus* (40%) *Staphylococcus* (20%) *Corynebacterium* (10%) *Lactobacillus* (10%) *Peptococcus* (10%)
Pei et al. [37]	2004	Broad-range 16S rDNA PCR	4	41	*Firmicutes* *Bacteroidetes* *Actinobacteria* *Fusobacteria* *TM7*	*Streptococcus* (39%) *Prevotella* (17%) *Veilonella* (14%)

Studies utilizing sequencing and selective media cultures demonstrate a predominantly Gram-positive microbiota inhabiting the normal esophagus. All studies found *Streptococcus* to be a dominant genus in normal esophageal microbiota. rDNA = ribosomal DNA; PCR = polymerase chain reaction; TM-7 = Saccharibacteria; rRNA = ribosomal ribonucleic acid; N/A = not available.

### 2.3. Microbiota Composition in Pathological Esophageal Tissue

The microbiota composition in pathological states, including GERD, BE, and EAC, differs from healthy esophageal tissue (Table 2). Deshpande et al. reported that GERD samples showed increased Gram-negative and decreased Gram-positive bacterial composition [9]. Similar results were observed in BE samples [9]. However, there is no consistent trend observed across the studies analyzing EAC samples. Elliott et al. found that EAC did not selectively express more Gram-negative microbiota, while Lopetuso et al. found that EAC was dominated by Gram-negative anaerobes [47,48].

Several studies evaluated the effect of the EAC cascade on microbial diversity. Both Elliott et al. and Snider et al. noted a decreased microbial diversity in EAC [47,49]. This observation may be due to an altered tumor environment in which only select microbes can survive. Lopetuso et al., on the other hand, noted an increased microbial diversity in BE and EAC compared to controls [48]. Furthermore, Deshpande et al. and Peter et al. noted no differences in microbial diversity between normal and pathologic samples [9,42]. Consequently, a consensus has not been reached regarding the effect of the EAC cascade on microbial diversity.

Lactic acid-producing bacteria have been shown to be in abundance in the EAC cascade microenvironment [9,50]. Increased lactic acid production as a result of this dysbiotic shift may contribute to the Warburg effect, as discussed below. Overall, however, future studies analyzing the microbiome in EAC samples in large sample populations are needed to understand the clinical impact of these findings.

**Table 2 biology-10-00697-t002:** Microbiota expressed by pathological esophagus.

Article	Publication Year	Method	Sample Size				Findings
			Control	GERD	BE	EAC	
Peter et al. [42]	2020	16S rRNA	12	N/A	31	10	Microbial diversity did not statistically differ between control and pathologic esophageal samples Control and pathologic tissue samples did not express statistically significant differences in microbial diversity Similar phyla were expressed in the oral cavity when compared to the esophagus Dominant genera, on average, in all esophageal samples included *Tissierella soehngenia* (16.67%), *Lactobacillus* (7.15%), *Streptococcus* (7.27%), *Acinetobacter* (5.80%), and *Prevotella* (5.24%) Planctomycetes and Crenarchaeota phyla were decreased in all pathologic samples when compared to control samples When compared to control groups, the following genera were downregulated in BE samples with high grade dysplasia: *Nitrosopumilus, Balneola,* and *Planctomyces*
Zhou et al. [50]	2020	16S rRNA	16	N/A	17	6	Increased relative abundance of Gram-negative *Fusobacteria* and *Proteobacteria* in BE Increased lactic acid-producing bacteria in EAC Decreased *Actiobacteria* and increased *Firmicutes* in EAC
Lopetuso et al. [48]	2020	16S rRNA	10	N/A	10	6	Control samples expressed increased Bacillus and Streptococcus Increased α- and β-diversity in BE and EAC Decreased *Streptococcus* in BE and EAC Increased *Prevotella*, *Veillonella*, and *Leptotrichia* in BE and EAC Progressive decrease in *Firmicutes:Bacteroidetes* ratio from BE to EAC
Snider et al. [49]	2019	16S rRNA	16	N/A	25	4	Decreased α-diversity in EAC High-grade-dysplasia BE and EAC expressed reduced *Firmicutes* and *Veillonella* High-grade-dysplasia BE and EAC expressed increased *Proteobacteria*, *Enterobacteriaceae, Akkermansia muciniphila*
Deshpande et al. [9]	2018	16S rRNA 18S rRNA	59	29	5	1	The esophageal microbiome clustered into functionally distinct community types defined by the relative abundances of *Streptococcus* and *Prevotella* Esophageal microbiome was enriched with Gram-negative oral-associated bacteria and microbial lactic acid production in early stages of the esophageal adenocarcinoma cascade (GERD and BE) Disease did not significantly alter alpha diversity measures of esophageal microbiome Taxa enriched in disease and not controls included *Leptotrichia, Fusobacterium, Rothia, Campylobacter, Capnocytophaga*
Elliott et al. [47]	2017	16S rRNA	16	N/A	17	15	EAC tissue had decreased microbial diversity compared to controls regardless of sampling location *Lactobacillus fermentum* was enriched in EAC samples (*p* = 0·028) Lactic acid bacteria dominated the microenvironment in 7/15 cases of EAC Decreased proportional abundance of genera in EAC included Gram-negative (*Veillonella, Megasphaera*, and *Campylobacter*) and Gram-positive taxa (*Granulicatella, Atopobium, Actinomyces,* and *Solobacterium*)
Gall et al. [51]	2015	16S rRNA	N/A	N/A	12	N/A	BE microbiota was dominated by *Streptococcus* and *Prevotella* species The ratio of *Streptococcus* to *Prevotella* independently correlated with waist-to-hip ratio in a positive manner and hiatal hernia length in a negative manner—two known risk factors for EAC in BE patients
Amir et al. [52]	2014	16S rRNA	15	N/A	6	N/A	No single taxon was found to be a distinguishing biomarker between normal esophageal tissue and abnormal esophageal tissue Increased levels of *Enterobacteriaceae* were observed in BE gastric fluid compared to controls
Yang et al. [46]	2009	16S rRNA	12	N/A	10	N/A	Esophageal microbiota can be classified into two groups The type 1 microbiome was predominantly composed of Gram-positive bacteria and was dominated by the *Streptococcus* genus Type 1 microbiome primarily correlated with normal esophagus The type 2 microbiome was predominantly composed of Gram-negative anaerobes/microaerophiles Type 2 microbiome primarily correlated with BE (odds ratio 16.5) Reduced amount of the *Streptococcus* genus in BE samples compared to control samples
Macfarlane et al. [53]	2007	16S rDNA	7	N/A	7	N/A	*Campylobacter* colonized the esophagus in 4/7 of BE patients *Campylobacter* was not identified in the control group

Results from nine studies that used 16s rRNA sequencing to analyze esophageal microbiota in controls, GERD, BE, and EAC. Results across studies are variable, which may be attributable to different sample collection methods and patient populations. Several studies show that GERD and BE express increased Gram-negative and decreased Gram-positive microbiota. According to two studies, EAC expresses decreased microbial diversity. A consensus regarding the microbiota composition in EAC has not been reached. Lactic acid-producing bacteria are increased in GERD, BE, and EAC. GERD = gastroesophageal reflux disease; BE = Barrett’s esophagus; EAC = esophageal adenocarcinoma; rDNA = ribosomal DNA; N/A = not available; rRNA = ribosomal ribonucleic acid.

### 2.4. Controversial Associations between Helicobacter pylori and the EAC Cascade

*Helicobacter pylori* (*H. pylori*) is not a dominant bacteria in normal or pathologic esophageal tissue. However, it plays an important but controversial role in the EAC cascade. It is a Gram-negative helical bacteria that is commonly present in the stomach. By utilizing catalase, urease, and oxidase enzymes, *H. pylori* is able to survive in the otherwise acidic environment of the stomach. Its association with malignancies such as gastric cancer and gastric mucosa-associated lymphoid tissue lymphoma has led to a plethora of studies investigating the relationship between *H. pylori* and the EAC cascade. 

In regard to GERD, a recent meta-analysis of randomized controlled trials conducted from 1990–2019 demonstrated that *H. pylori* eradication increased the risk of new erosive GERD by two-fold [54]. This finding was not consistent across all studies, however. Contrastingly, a meta-analysis by Yaghoobi et al. demonstrated no association between *H. pylori* eradication and the risk of new GERD development [55]. Similar controversies have been raised regarding the relationship between *H. pylori* and preexisting GERD. Saad et al.’s meta-analysis demonstrated an improvement in GERD symptoms with *H. pylori* eradication, while Zhao et al.’s findings suggest that *H. pylori* does not have a significant effect on the healing or relapse rate of preexisting GERD [54,56]. It is evident from the literature review that the role of *H. pylori* in new GERD development as well as improvement following its eradication has been controversial and requires further research. 

The effect of *H. pylori* on BE is also controversial. A meta-analysis of 72 studies composed of 84,717 total patients with BE found that *H. pylori* infection was associated with a reduced risk of dysplastic, non-dysplastic, and long-segment BE [57]. Similar results were reported in a case-control study where BE was associated with a lower prevalence rate of *H. pylori* infection when compared to control [58]. This association was recently analyzed in a cohort study of 81,919 patients undergoing eradication treatment for *H. pylori* [59]. Despite the previously established association between *H. pylori* and BE, the cohort study did not provide evidence of an increased risk of BE following *H. pylori* eradication. Consequently, future studies are essential in elucidating whether the relationship between *H. pylori* and BE is merely an association or a more established mediator of BE pathophysiology. 

Recent studies have also evaluated *H. pylori* in association with EAC. A meta-analysis of 28 studies discerned that *H. pylori* infection is inversely associated with EAC [60]. Cytotoxin-associated gene A strains specifically were less likely to be associated with EAC when compared to strains negative for cytotoxin-associated gene A. Conversely, a literature review by Polyzos et al. remarks that meta-analysis of observational studies suggests an inverse association between *H. pylori* infection and EAC, whereas epidemiologic studies are collectively inconclusive [61].

Overall, the lack of consensus regarding the implications of the associations between *H. pylori* and the EAC warrants future studies.

## 3. Role of Microbiota in the Pathogenesis of the EAC Cascade

The dysbiotic shift in esophageal microbiota during the EAC cascade may promote the pathogenesis of EAC through five possible mechanisms: (i) activation of toll-like receptors, (ii) stimulation of cyclooxygenase-2 expression and subsequent delayed gastric emptying, (iii) stimulation of iNOS expression leading to relaxation of the lower esophageal sphincter, (iv) stimulation of the NLRP3 inflammasome, and (v) increased lactate availability for the Warburg effect (Figure 1).

### 3.1. Microbial Activation of Toll-Like Receptors

The observed dysbiotic microbiota may aberrantly activate TLRs and lead to NF-κB, CREB, AP-1, and IRF activity that promotes EAC cascade development. 

#### 3.1.1. Background 

Toll-like receptors (TLRs) are pattern recognition receptors (PRRs) expressed by immune and epithelial cells. As a component of the innate immune system, TLRs lead to phagocytosis, inflammatory cytokine release, and complement activation [62,63,64,65]. TLRs also play an essential role in linking the innate and adaptive immune systems due to their presence on antigen-presenting cells [66]. TLRs consequently play an essential role in the gastrointestinal tract where a delicate balance between immunity against pathogens and tolerance of symbiotic bacteria must be maintained. TLRs differentiate between pathogenic and symbiotic microbiota by recognizing pathogen-associated molecular patterns (PAMPs), unique conserved structures native to both pathogenic and non-pathogenic microorganisms, or damage-associated molecular patterns (DAMPs). PAMPs expressed in gastrointestinal tract microbiota include Gram-negative bacterial lipopolysaccharide (LPS), bacterial peptidoglycan, specific bacterial RNA/DNA characteristics, as well as many others. When TLRs recognize PAMPs or DAMPs, there is a downstream activation of transcription factors that regulate cytokine gene expression (NF-κB, AP-1, CREB, IRFs, IFN-α/β); proliferation, differentiation, and apoptosis (AP-1); and cellular mechanisms involved in carcinogenesis (IFN-α/β) (Table 3).

Considering the role of TLRs in immunity as well as their downstream transcription factors, TLRs are capable of being anti-tumorigenic or pro-tumorigenic. Whether the TLR is anti- or pro-tumorigenic depends on the TLR, cancer subtype, and the immune cells infiltrating the tumor [68]. TLRs can be pro-tumorigenic through pro-inflammatory cytokines, anti-apoptotic signaling, proliferative signaling, and profibrogenic signals in the tumor microenvironment or tumor cells themselves [69]. This pro-tumorigenic role of TLRs has been demonstrated in colitis-associated cancer, where TLR recognition of microbiota promoted the development of invasive carcinoma [70]. On the other hand, TLR stimulation can lead to anti-tumorigenic effects through T-cell-mediated immunity and recognition of tumor-associated DAMPs [68,71]. This anti-tumorigenic effect is demonstrated in several cancers, as summarized by Dajon et al., but has yet to be demonstrated in EAC [68]. Considering both the anti-tumorigenic and pro-tumorigenic roles of TLRs, the potential activation of TLRs by dysbiotic esophageal microbiota in the EAC cascade is of significant interest. 

#### 3.1.2. TLR1, TLR2, and TLR6

The role of TLRs, including TLR2, has been studied in the EAC cascade. Mulder et al. studied TLR expression in esophageal cell lines. Their results demonstrated TLR2 expression only in EAC cell lines and not normal cell lines [72]. However, Verbeek et al. report TLR2 mRNA expression in inflammatory cells as well as epithelial cells in biopsies from patients with normal esophagus, reflux esophagitis, BE, and EAC [73]. Furthermore, TLR2 expression was increased in reflux esophagitis, BE, and EAC relative to normal esophageal epithelium [73]. Similar findings in regard to TLR2 expression in the EAC cascade have been demonstrated in additional studies [74,75]. Collectively, these studies show that TLR2 is overexpressed in inflammatory states like GERD, BE, and EAC. 

Studies also demonstrate the variable location of TLR2 expression along the EAC cascade. TLR2 was expressed in basal keratinocytes of normal esophageal epithelium and in superficial epithelial cells, crypts, and lamina propria in BE [73]. EAC, on the other hand, demonstrated diffuse TLR2 expression throughout the biopsy [73]. This observation raises the question of whether progressive mucosal disruption in the EAC cascade exposes epithelial TLR2 to PAMPs of dysbiotic microbiota not tolerated by the immune system. 

TLR2 may play a heightened role in dysbiotic microbial recognition along the pathogenesis of the EAC cascade due to its ability to heterodimerize with other TLRs and recognize a wider variety of ligands. Current research demonstrates that TLR2/6 and TLR1/6 heterodimers recognize several PAMPs, including components of bacterial cell walls known as diacylated and triacylated lipoproteins [74,76]. Heterodimerization, specifically in the EAC cascade, was suggested by Huhta et al.’s immunohistochemistry analysis of TLR1, TLR2, and TLR6 expression in pathologic esophageal samples [74]. Study results demonstrated a stepwise increase in TLR1 and TLR6 from normal esophageal tissue to high-grade dysplasia. The increased expression of TLR2, TLR1, and TLR6 along the EAC cascade, may consequently enable TLR2 to heterodimerize and recognize a greater diversity of dysbiotic PAMPs. Additionally, the microbiota itself may also mediate TLR2 expression. Hörmann et al. demonstrated a microbiota-dependent upregulation of TLR2 as well as TLR1 in small intestinal mucosa [77]. Even colonization with single microbes was capable of eliciting this response. This effect was reversed with a seven-day course of broad-spectrum antibiotics [77]. Consequently, TLR2 mediates microbiota, and microbiota mediate TLR2. Studies evaluating this relationship in the EAC cascade specifically are warranted. 

In addition to the abovementioned heterodimerization and pro-inflammatory signaling, TLR2 can regulate esophageal epithelial barrier function and mediate proliferation of epithelial cells. Specifically, TLR2 activation with bacterial PAMP peptidoglycan and fungal zymosan in normal esophageal epithelium leads to upregulation of tight junction complexes claudin-1 and zonula occludens-1, strengthening esophageal epithelial barrier function [78]. This homeostatic regulation of esophageal epithelial barrier function by TLR2 mediates the ability of microbes and DAMPs to pass through the mucosal barrier and further activate the host immune system [78]. Consequently, TLR2 may be an essential protective mediator of the EAC cascade in which the microbial shifts and reflux-mediated damage lead to aberrant expression of PAMPs and DAMPs. The effect of TLR2 on epithelial cell proliferation was demonstrated by Hörmann et al., who showed that TLR2 agonists significantly increased proliferation of small intestinal epithelial cell lines through multiple protein kinase pathways and moderately increased apoptosis [77]. The effect of TLR2 on epithelial proliferation was further supported by decreased proliferation in TLR2 deficient mice [77]. If TRL2 elicits the same effect in the esophagus, the interplay between microbiota and TLR2 may affect the propensity for mucosal damage and tumorigenesis in the EAC cascade. 

In summary, TLR2 overexpression and potential for heterodimerization, pro-inflammatory signaling, and epithelial barrier regulation in the EAC cascade have been established. Currently, there is a paucity of studies analyzing how TLR2 expression directly varies with composition of the esophageal microbiota, although TLR2 expression in the small intestine has been shown to be regulated by microbiota. Further studies examining TLR2 in relation to normal and dysbiotic esophageal microbiota are warranted. Elucidating this relationship would help establish whether the dysbiotic activation of TLR2 in the EAC cascade triggers a pro-inflammatory cascade or a protective regulation of the esophageal epithelial barrier. 

#### 3.1.3. TLR 4

TLR4 is perhaps the most well-studied TLR in relation to the EAC cascade. Kohtz et al. determined how reflux conditions, as seen in GERD, affect TLR4 expression [79]. Their analysis demonstrated that TLR4 expression increased in normal cell lines under reflux conditions simulated by acidic or bile-containing solutions [79]. Similarly, Verbeek et al. demonstrated increased TLR4 expression in EAC (3.2 folds), BE (2.7 folds), and reflux esophagitis (1.9 folds) compared to normal squamous epithelium esophageal samples [80]. The location of TLR4 expression also differed along the EAC cascade. Specifically, TLR4 expression in normal squamous epithelium samples was mainly confined to the basal layer of the squamous epithelium. In contrast, TLR4 expression in BE samples was observed in superficial epithelial cells, crypts, and lamina propria cells. EAC expression of TLR4 was diffusely distributed across superficial and infiltrating cells. Huhta et al.’s findings support Verbeek et al.’s observations regarding increased TLR4 expression in abnormal esophageal samples and elaborate that TLR4 expressed by EAC localizes in the nucleus rather than the normally observed cytoplasm [74,80]. This nuclear localization of TLR4 in EAC was found to correlate with distant metastasis and poor prognosis. 

Increased expression of TLR4 in pathological states of the EAC cascade has severe implications for the role of microbiota in EAC development [74,79,80]. When TLR4 is normally confined to the basal layer in the squamous epithelium of the esophagus, it may not come into contact with PAMPs expressed by esophageal microbiota. However, gastric reflux conditions may lead to an increase in TLR4 expression and damage the superficial esophageal epithelium normally separating microbial PAMPS from basally expressed TLR4. Consequently, the reflux conditions seen in the EAC cascade may not only increase the expression of TLR4 but also expose TLR4 to microbial PAMPs. 

The implications of TLR4 activation by microbial PAMPs in this manner were elucidated by Verbeek et al. [80]. Analysis was performed with ex vivo cultures of BE and normal squamous esophagus biopsies as well as non-neoplastic BE cell lines. Downstream effects of TLR4 activation by LPS were monitored by NF-κB, IL-8, and COX-2 expression analysis with western blots, ELISA, or quantitative RT PCR, respectively. Results demonstrated that LPS stimulation of TLR4 caused: (i) NF-κB activation; (ii) a dose-dependent increase in IL-8 expression and secretion in non-neoplastic BE cell lines and ex vivo cultures of squamous esophagus, BE, and duodenum biopsies; and (iii) increased COX-2 expression in ex vivo BE biopsies and BE cell lines, but not ex vivo cultures of normal squamous esophagus or duodenum. Kohtz et al. noted a similar effect of TLR4 activation by LPS in normal and EAC cell lines [79]. Specifically, LPS-activation of TLR4 led to downstream NF-κB activation and increased esophageal cell growth rates. Of note, inhibiting NF-κB attenuated the accelerated growth rates previously observed [79].

These results suggest that LPS activation of TLR4 can lead to activation of NF-κB, inflammation, and cell proliferation. Increased expression of Gram-negative bacteria, known to uniquely express LPS, are increased in the EAC cascade. Therefore, the shift in EAC cascade microbiota may enhance the probability of LPS activating TLR4 and lead to consequent activation of NF-κB, secretion of IL-8, increased COX-2 expression, and increased proliferation [80]. Each of these downstream effects may promote carcinogenesis. NF-κB, as discussed previously, is a key regulator of inflammation, proliferation, and cell survival. COX-2 can mediate cell proliferation, apoptosis inhibition, angiogenesis, tumor invasiveness, and immunosuppression, as discussed below [81]. The role of IL-8 in carcinogenesis includes regulation of angiogenesis, cancer cell growth and survival, tumor cell motion, leukocyte infiltration, and immune responses [81]. Consequently, the Gram-negative shift in BE microbiota may enable the carcinogenesis of EAC cascade through the activation of TLR4. 

#### 3.1.4. TLR 5

Helminen et al. assessed the expression of TLR5 in normal esophageal squamous epithelium, BE with and without dysplasia, and EAC biopsies by means of immunohistochemistry staining [82]. TLR5 expression was only noted in the basal third of normal squamous epithelium and non-dysplastic BE tissue. In contrast, dysplastic BE and EAC tissue demonstrated increased, diffuse TLR5 expression with no apparent polarization. Similar to Verbeek et al.’s analysis of TLR4, this suggests that esophageal epithelium damage and transformation may expose TLR5 to microbial PAMPs and lead to consequent inflammatory cascades [80].

The composition of the dysbiotic microbiota in the EAC cascade further increases the likelihood of TLR5 activation, considering Snider et al. noted an increased relative abundance of the Proteobacteria phylum in high-grade dysplasia BE and EAC [49]. Proteobacteria is a phylum predominantly composed of bacteria with flagella, a well-established PAMP for TLR5 [83]. If this dysbiotic shift triggers TLR5, it can lead to downstream inflammatory cascades via NF-κB, CREB, and AP-1. 

The potential for inflammation downstream of TLR5 may make TLR5 an important marker for EAC cascade progression. Specifically, Helminen et al. found moderate to high expression of TLR5 to be a marker for low-grade dysplasia with 86% sensitivity and 83% specificity [82]. Consequently, TLR5 may serve as a diagnostic marker for esophageal columnar dysplasia and BE, a condition known for controversial diagnostic parameters. This would have positive implications for prophylactic EAC treatment and monitoring of BE. However, a study by Helminen et al. did not find TLR5 expression to correlate with EAC prognosis [82].

Although TLR5 expression, the potential for activation by a dysbiotic flagellated microbiota, and correlation with dysplasia have been elucidated, more studies are needed to identify the direct interactions between dysbiotic microbiota and TLR5. Analyzing how TLR5 expression relates to the relative abundances of dysbiotic flagellated bacteria would provide the foundation on which additional studies could analyze this relationship. 

#### 3.1.5. TLR 9

TLR9 expression in EAC was assessed by Kauppila et al. by means of immunohistochemical staining [84]. TLR9 expression was observed in all 76 EAC tumors. Furthermore, strong cytoplasmic TLR9 immunoreactivity correlated with high pathological tumor stage, positive lymph nodes, distant organ metastases, high tumor grade, tumor unresectability, and decreased 10-year survival rates. Kauppila et al.’s findings lay the foundation for future studies investigating the role of TLR9 in EAC. Specifically, while the correlation between TLR9 and tumorigenesis was determined, the downstream effects in EAC carcinogenesis were not determined. Future studies will need to analyze how TLR9 directly or indirectly mediates tumorigenesis in the EAC cascade. 

Furthermore, the ligand for TLR9 in EAC was not determined. Previous studies have shown that TLR9 can recognize CpG DNA sequences present in bacteria and viruses [67,85]. The viral composition of the esophageal microbiota has yet to be fully elucidated. However, bacterial unmethylated CpG DNA sequences in the esophageal microbiota could be responsible for the observed correlations between TLR9 expression and EAC tumorigenesis. Studies analyzing the effects of dysbiotic CpG DNA expression in relation to TLR9 expression in the EAC cascade would consequently be of interest. 

#### 3.1.6. Summary of TLRs in the EAC Cascade 

TLR2 expression is increased in the EAC cascade, and the location of TLR2 expression varies with the pathologic state. It is capable of heterodimerizing with TLR1 and TLR6 and regulating esophageal epithelial barrier function. 

TLR4 expression is increased in a stepwise fashion along the EAC cascade, and the location of TLR4 expression varies with the pathologic state. After recognizing the Gram-negative PAMP LPS, it causes pro-inflammatory signaling through NF-κB, IL-8, and COX-2. 

TLR5 expression is increased in BE and EAC, and the location of TLR5 expression changes depending on the pathologic state. TLR5 expression has been shown to be a marker for low-grade esophageal dysplasia. 

TLR9 is expressed in EAC tumors, and cytoplasmic TLR9 immunoreactivity correlates with high pathological tumor stage, positive lymph nodes, distant organ metastases, high tumor grade, tumor unresectability, and decreased 10-year survival rates [84].

### 3.2. Microbial Stimulation of Cyclooxygenase-2 Expression

The Gram-negative, dysbiotic microbiota observed in the EAC cascade may stimulate COX-2 overexpression that promotes GERD development and neoplastic progression of BE into EAC. 

Cyclooxygenase-2 (COX-2) is a rate-limiting enzyme that catalyzes the initial step of arachidonic acid metabolism into prostaglandin H2, a precursor to prostanoids such as prostaglandin, thromboxane, and prostacyclin. These products act as autocrine and paracrine lipid mediators that maintain local homeostasis by mediating vascular function, wound healing, and inflammation [86,87]. COX-2 expression is initiated in response to IL-1, IFN-γ, TNF-α, and epidermal growth factor receptor [88]. COX-2 has substantial consequences in cancers, considering it is capable of mediating cell proliferation, apoptosis inhibition, angiogenesis, invasiveness, and immunosuppression [89]. Overexpression of COX-2 has been shown to induce tumorigenesis in mammary epithelium, and COX-2/prostaglandin E2 dysregulation can promote colorectal tumorigenesis through tumor maintenance, metastatic spread, and perhaps tumor initiation [86,90]. Several studies have already demonstrated increased COX-2 protein expression in esophageal samples of patients with BE and adenocarcinoma compared to the normal esophagus; however, the inciting factor triggering increased expression of COX-2 is not well defined [88,91,92]. 

The dysbiotic shift towards a higher relative abundance of Gram-negative bacteria in the EAC cascade may be a source of this increased COX-2 expression. Verbeek et al. demonstrated that ex vivo cultured BE biopsies incubated with LPS, a PAMP expressed by Gram-negative bacteria, had increased expression of COX-2 downstream of TLR4 [80]. This suggests that LPS may be the mediator through which COX-2 expression is increased.

Increased COX-2 expression as a result of LPS activation may have several effects on the EAC cascade. First, dysbiotic LPS-induced COX-2 expression may mediate the development of GERD. Building on Collares’ observation, Calatayud et al. sought to determine the mechanism by which LPS induced a delay in gastric emptying, a possible GERD risk factor [93,94]. Rats were treated with LPS as well as different prostaglandin and COX-2 inhibitors. Results demonstrated that dual COX-1 and COX-2 inhibitors prevented the LPS-induced delay in gastric emptying to a similar degree as COX-2 inhibitors alone. Phospholipase-2 inhibitors did not modify the rate of gastric emptying in LPS-treated animals. These results suggest that LPS stimulates the release of prostanoids and consequently delay gastric emptying through COX-2 rather than phospholipase expression. Delayed gastric emptying distends the stomach and increases the quantity of food available for reflux into the esophagus. This also generates transient lower esophageal sphincter relaxations that may facilitate reflux into the esophagus [95]. However, other studies failed to show similar results and a consensus behind the mechanism of gastroparesis is still not well established [96,97,98,99].

The LPS-induced COX-2 expression is also implicated in the potentiation of tumorigenesis. COX-2 overexpression has been shown to be a key mediator of BE neoplastic progression into EAC by promoting angiogenesis, cell proliferation, and reduced apoptosis [100]. Morris et al. specifically noted that COX-2 expression increased from low-grade dysplasia to high-grade dysplasia in the esophagus [88]. This result implies that COX-2 expression may contribute to cancer development rather than cancer inducing COX-2 expression. Furthermore, meta-analysis shows that overexpression of COX-2 is associated with poor overall survival, depth of invasion, metastasis, and tumor, node, and metastasis (TNM) stage in esophageal cancers [101].

In summary, LPS, expressed by the dysbiotic shift to a Gram-negative microbiota in the EAC cascade, leads to an increase in COX-2 expression. Increased COX-2 expression, in turn, leads to delayed gastric emptying that may increase the risk of GERD. Furthermore, COX-2 expression may promote carcinogenesis by promoting angiogenesis, cell proliferation, and a reduction in apoptosis. With these findings as a foundation, more studies are needed to elucidate the direct relationship between COX-2 and the EAC cascade microbiota. 

### 3.3. Microbial Stimulation of Inducible Nitric Oxide Synthase 

iNOS stimulated by Gram-negative, dysbiotic microbiota observed in the EAC cascade may relax the lower esophageal sphincter and promote GERD development.

Inducible nitric oxide synthase (iNOS) is an enzyme that produces nitric oxide (NO) by oxidizing L-arginine. Although iNOS is predominantly expressed by macrophages, iNOS expression can be induced in nearly any cell type with the appropriate stimulatory factors such as bacterial LPS and cytokines. NO produced by iNOS is capable of damaging both pathogenic cells and host cells through several mechanisms. Specifically, NO has a large affinity for protein-bound iron and is consequently capable of blocking iron-dependent enzymes that play an essential role in the citric acid cycle, mitochondrial electron transport, and DNA replication [102]. High concentrations of NO can also form inflammatory radicals, directly causing single and double-stranded DNA breaks, and inhibiting DNA repair enzymes through tyrosine and cysteine nitrosation [102,103]. iNOS activation also enables radical damage through peroxynitrite, a radical formed by NO interaction with superoxide [103,104].

Each of these mechanisms can have a protective effect on the host by killing tumor cells and microorganisms. However, similar effects can be directed against host cells, causing inflammation and tissue damage. iNOS activity and consequent high concentrations of NO have been shown to be involved in several cytotoxic processes such as apoptosis, angiogenesis, and DNA damage during cancer development [104]. Consequently, iNOS can play both an anti-tumorigenic and pro-tumorigenic role in cancers. There is reason to suspect iNOS in the pathogenesis of the EAC cascade considering several studies have demonstrated that iNOS is overexpressed in BE and EAC compared to normal esophagus [91,105,106]. 

The role of microbiota in the observed iNOS overexpression has been evaluated by several studies. Specifically, several studies demonstrated that LPS caused a dose-dependent decrease in lower esophageal sphincter tone, an effect attenuated by selective inhibition of iNOS with aminoguanidine or L-canavanine [107,108]. Park et al. evaluated this relationship further and analyzed these changes directly related to iNOS expression [109]. The study analyzed plasma nitrite/nitrate levels and iNOS expression in opossum models by western blot and RT-PCR assays before and after LPS administration. Results showed that LPS treatment increased plasma nitrite/nitrate levels and iNOS expression in the lower esophageal sphincter. To establish the role of iNOS in this observation, a selective iNOS inhibitor was administered. iNOS inhibitor treatment significantly attenuated the increase in plasma nitrite/nitrate levels and slightly attenuated the increased iNOS expression. Consequently, LPS expressed by Gram-negative dysbiotic microbiota along the EAC cascade can induce increased iNOS expression in the lower esophageal sphincter and subsequently causing lower esophageal sphincter relaxation. 

Relaxation of the lower esophageal sphincter in a chronic fashion enables the development of GERD, a known risk factor for EAC. 

### 3.4. NLRP3 Inflammasome Activation by Microbiota 

Dysbiotic microbiota in the EAC cascade may trigger NLRP3 Inflammasomes independently of TLRs. 

Inflammasomes are expressed by epithelial and immune cells. Inflammasomes interact with receptors, including TLRs and nod-like receptors (NLR)s, to monitor damage or foreign microorganisms, and trigger an immune response when appropriate. The nod-like receptor protein 3 (NLRP3) inflammasome has specifically been shown to be modulated by gastrointestinal dysbiosis but also regulates gastrointestinal dysbiosis [110]. Activation of NLRP3 requires two steps. First, there is a priming signal in which microbial ligands activate TLRs or cytokines activate tumor necrosis factor receptors (TNFRs). Either the activated TLR or TNFR causes activation of NF-κB, leading to upregulation of pro-IL-1β and NLRP3 transcription [111]. A second signal, stimulated by DAMPs or PAMPs such as LPS, then promotes the assembly of the inflammasome complex, causing the activation of the NLRP3 inflammasome [111].

Activation of the NLRP3 inflammasome in this manner may regulate the composition of the esophageal microbiome. Yao et al. demonstrated that hyperactive NLRP3 inflammasomes improve the symbiotic relationship between gut microbiota and the immune system through increased regulatory T cell induction [112]. The influential role of the NLRP3 inflammasome in microbiota composition was also demonstrated by Hirota et al. [113]. Specifically, the study demonstrated impaired crypt bactericidal activity in NLRP3 knock-out mice with a subsequent dramatic change in intestinal microbiota. NLRP3 knock-out mice were also associated with decreased innate immune mechanisms combating mucosal injury by intestinal microbiota [113]. This suggests that NLRP3 may play an important protective role in maintaining symbiosis between host immune systems and gut microbiota. Li et al. alternatively demonstrated that NLRP3 activation in mice models of acute pancreatitis plays less of a regulatory role and instead disrupts gut microbiota through NLRP3 induced inflammation [114]. Collectively, these studies suggest that NLRP3 plays an influential role in gut microbiome composition and microbiota symbiosis with the host. Whether NLRP3 activation promotes symbiosis or dysbiosis in the EAC cascade will require additional studies analyzing the esophageal microbiome specifically. 

On the other hand, the impact of dysbiosis on NLRP3 inflammasome activation has been evaluated in regard to the EAC cascade. Considering LPS is both an activating molecule for NLRP3 inflammasome and a PAMP with increased expression in the EAC cascade microbiota, Nadatani et al. analyzed NLRP3 in regard to LPS and the EAC cascade [115]. Normal esophageal squamous cells and BE epithelial cells were treated with LPS, and the following were measured with and without TLR4 or NLRP3 inflammasome inhibition: (i) TLR4, pro-IL1β, pro-IL18, and NLRP3 expression; (ii) caspase-1 activity; (iii) TNF-α, IL8, IL1β, and IL18 secretion; (iv) lactate dehydrogenase release (a pyroptosis marker); and (v) mitochondrial reactive oxygen species (ROS). Results demonstrated that LPS both primes and activates the NLRP3 inflammasome via two separate mechanisms. First, LPS primes for NLRP3 inflammasome activation by inducing the expression of NLRP3, pro-IL1β, and pro-IL18 downstream of TLR4. It is important to note that LPS only has this priming effect downstream of TLR4 in BE cells, not normal esophageal squamous cells. Second, independently of TLR4, LPS increases mitochondrial ROS that activate the NLRP3 inflammasome. The activated NLRP3 inflammasome subsequently enables pyroptosis and converts pro-IL1β and pro-IL18 into mature IL1β and IL18. Therefore, LPS expressed by the Gram-negative dysbiotic microbiota in the EAC cascade can both prime and activate the NLRP3 inflammasome and lead to downstream production of IL-1β and IL-18.

Activation of the NLRP3 inflammasome in this manner may have significant consequences on the development of the EAC cascade, considering NLRP3 can directly promote or suppress tumorigenesis through IL-1β and IL-18 [116,117]. Depending on the cancer, IL-1 β and IL-18 have been shown to have pro-tumorigenic roles, including angiogenesis as well as anti-tumorigenic roles through immune cell modulation [118]. Whether the pro-tumorigenic or anti-tumorigenic pathway of NLRP3 predominates in the EAC cascade has yet to be determined. Recent studies demonstrate that increased NLRP3 expression in ESCC was associated with IL-1β as well as increased tumor proliferation, migration, and invasion [119]. Similar studies will need to be conducted in regard to the EAC cascade and their relationship with microbiota.

### 3.5. Microbial Contribution to the Warburg Effect 

Dysbiotic microbiota of the EAC cascade may increase lactate availability for EAC carcinogenesis.

Lactate is an essential metabolite and signaling molecule. It is needed for carcinogenic angiogenesis, immune system evasion, cell migration, and metastasis [120]. Lactate is particularly important for cancer metabolism in which there are increased metabolic demands. In esophageal cancers specifically, there is a marked reduction of blood glucose levels with an increased level of lactate compared to healthy controls [121]. This reflects the “Warburg effect”, in which cancers have increased glucose uptake and fermentation of glucose into lactate, enabling increased tumor survival and proliferation. 

Microbiota in the EAC cascade may facilitate the development of the Warburg effect. Specifically, Deshpande et al. analyzed microbiota in 106 microbial brush samples of normal, GERD, or BE esophagus [9]. Relative to normal esophagus brush samples, GERD and BE microbiota demonstrated an overall increase in lactic acid production. Similarly, Zhou et al. observed an EAC dysbiotic microbiota composed of a high abundance of lactic acid-producing bacteria, including Staphylococcus, Lactobacillus, Bifidobacterium, and Streptococcus [50]. These findings are significant, considering increased lactate availability may potentiate the Warburg effect in the EAC cascade. 

The Warburg effect is promoted by mutations that favor increased glucose uptake; increased glycolytic enzyme expression; increased lactate production, accumulation, and release; and upregulated lactate exchange between cells via monocarboxylate transporters (MCT)s. [120]. Specifically, in the EAC cascade, Huhta et al. demonstrated a linearly increasing cytoplasmic MCT1 and MCT4 expression from normal epithelium to BE to dysplasia and EAC [122]. Consequently, not only do dysbiotic microbiota in the EAC cascade increase lactate availability, but the EAC cascade expresses increased concentrations of MCTs capable of transporting lactate into the cell for utilization. Zhang et al. further supported this finding by analyzing serum metabolite profiles of patients with normal esophagus, BE, high-grade dysplasia, and EAC by LC-MS and NMR assays [123]. Results demonstrated that lactate was present at statistically increased levels in BE, high-grade dysplasia, and EAC compared to normal controls. 

Whether the increased relative abundance of lactate-producing bacteria is a cause or symptom of EAC is unclear at this point in time, however. Many patients with EAC experience dysphagia which limits their diet to a primarily liquid-dairy diet. Considering diet mediates microbial composition, EAC’s altered dairy-based diet may promote this dysbiotic shift to increased relative abundance of lactate-producing microbiota. 

In summary, dysbiotic microbiota in GERD, BE, and EAC are composed of increased relative amounts of lactate-producing bacteria. With increased MCT1 and MCT4 expression along the EAC cascade, the lactate produced by the dysbiotic microbiota can be transported into the cell for the Warburg effect and subsequent tumorigenesis. However, further studies are required to evaluate the role of antibiotics targeting the lactate-producing microbiota in the EAC cascade development. By doing so, the degree to which dysbiotic lactate production contributes to the development of the EAC cascade could be determined. Furthermore, tracing how the dysbiotic lactate products are used by normal, GERD, BE, and EAC cells would be of interest. 

## 4. Therapeutic Potential of Esophageal Microbiota 

As discussed above, many studies have evaluated the pathologic role of microbiota in EAC cascade pathogenesis. Recent studies are beginning to evaluate the esophageal microbiota from an alternative therapeutic perspective. Specifically, they are evaluating how supplementing the esophageal microbiome, rather than targeting it with antibiotics, may serve a therapeutic role in the EAC cascade. 

Prebiotics are compounds that stimulate growth or activity of microbiota and may be a mediator through which the esophageal microbiota can play a therapeutic role in the EAC cascade. A study by Selling et al. evaluated the effect of a prebiotic for Gram-positive lactobacilli in chronic GERD [124]. Daily supplementation of maltosyl-isomaltooligosaccharides as a prebiotic for lactobacilli led to a significant improvement of symptoms in patients with chronic GERD [124]. A systematic review suggested probiotics led to a similar improvement in GERD symptoms [125]. However, it has been noted that studies with improved internal validity are warranted to confirm the efficacy of probiotics in GERD [125].

Another mechanism through which esophageal microbiota may play a therapeutic role in the EAC cascade is through the bioengineered *Escherichia coli* Nissle 1917 [126]. It is capable of acting as a probiotic as well as producing and delivering antitumor treatment. In colitis, *Escherichia coli* Nissle 1917 has been shown to positively affect microbiota balance and induce and maintain ulcerative colitis remission [127]. The potential therapeutic effect of *Escherichia coli* Nissle 1917 has not been analyzed in the EAC cascade at this point in time, however it certainly can be a target for future research [126]. 

## 5. Conclusions and Future Directions

In summary, the EAC cascade expresses dysbiotic microbiota relative to the normal esophagus. The composition of microbiota shifts from a predominantly Gram-positive to a predominantly Gram-negative microbiota in GERD and BE. Although similar findings have been found in EAC, there are conflicting results, and a consensus has not been reached. LPS, a PAMP expressed by Gram-negative bacteria, may promote the development of the EAC cascade through (i) activation of TLRs; (ii) stimulation of COX-2 expression and subsequent delayed gastric emptying; (iii) stimulation of iNOS expression leading to relaxation of the lower esophageal sphincter; and (iv) priming and activation of the NLRP3 inflammasome. Dysbiotic microbiota in the EAC cascade also express an increased relative abundance of bacteria that utilize lactate metabolism. This increased availability of lactate may potentiate the Warburg effect and subsequent tumorigenesis. 

Amidst these findings, several key gaps in knowledge remain. Although a trend from a Gram-positive to a Gram-negative microbiota is noted in GERD and BE, studies have not reached a consensus in EAC. Furthermore, a consensus has not been reached regarding the specific bacterial species with increased expression along the EAC cascade. A portion of this variance is attributable to differences in host age, genetics, geography, environmental exposures, and diet. Consequently, additional studies investigating the composition of EAC cascade microbiota with larger sample sizes of a more diverse population are required. Another source of this variability is the diversity of microbiome analysis techniques. No consensus has been reached in regard to the most accurate technique for microbiome analysis. Furthermore, the same techniques for genomic analysis can vary with the method used to collect the sample. For example, the reproducibility of high-throughput sequencing of 16S rRNA gene amplicons is significantly dependent on the DNA extraction method [128]. Additionally, DNA extraction for high-throughput sequencing is not selective for DNA from live bacteria. DNA from dead bacteria may also be extracted from the sample. It is consequently essential to have standardized microbiome sample collection methods and analysis techniques for future studies. Similarly, standardized methods are needed to further evaluate the association and implications of *H. pylori* in the EAC cascade if a consensus is to be reached. 

The relationship between dysbiotic microbiota, TLRs, and the EAC cascade also warrants additional studies. Although TLRs are known to interact with PAMPs expressed by dysbiotic microbiota such as LPS, the degree to which PAMPs expressed by dysbiotic microbiota directly activate TLRs and mediate the EAC cascade remains unknown. Future studies utilizing TLR receptor modifiers among dysbiotic microbiota in the EAC cascade would greatly advance our understanding of how dysbiotic microbiota interact with TLRs and mediate the development of the EAC cascade. Considering TLR4 recognizes LPS, a PAMP with increased expression along the EAC cascade, studies investigating the direct effects of TLR4 on the EAC cascade would be of particular interest. In contrast to TLR4, analysis of TLR2 in relation to the microbiota would focus on the protective effects downstream of TLR2 and whether dysbiotic microbiota promote or interfere with TLR2 regulation of epithelial tight junctions. 

Microbial activation of COX-2 and iNOS may separately promote GERD development. Although LPS has been established as an activator for both, whether the observed shift towards a Gram-negative microbiota can directly activate their expression and subsequent GERD development remains unknown. Consequently, how the expression of each varies in relation to dysbiotic microbial composition warrants investigation.

Analysis of whether LPS priming and activation of the NLRP3 inflammasome protects against or promotes EAC carcinogenesis is also needed. The effects of the NLRP3 inflammasome on tumorigenesis have been shown to vary from cancer to cancer. It will be important to elucidate whether it promotes or inhibits EAC development. Furthermore, how the effect of the NLRP3 inflammasome directly mediates the esophageal microbiome in the EAC cascade and how dysbiosis manipulates NLRP3 inflammasome function have yet to be determined. Further studies investigating the role of the esophageal microbiome in EAC pathogenesis and the changed molecular mechanism leading to EAC are warranted. Investigating the cause of change in the esophageal microbiome leading to EAC will facilitate in-depth understanding and prophylactic strategies to decrease EAC incidence.

Considering both lactate-producing bacteria and MCTs are increased in GERD, BE, and EAC, how the microbial lactate products are utilized by pathologic cells in the EAC cascade is of extreme importance. If the lactate is utilized for the Warburg effect in EAC, lactate and the dysbiotic microbiota producing it may be important targets for slowing EAC tumorigenesis.

Finally, the therapeutic role of microbiota in EAC pathogenesis is a new approach through which there are many avenues for future studies. Prebiotics and probiotics show promising beginnings, but more studies with strong internal validity are needed to evaluate the potential therapeutic role of prebiotics and probiotics in the EAC cascade. Additionally, *Escherichia coli* Nissle 1917 is a novel microbial mechanism through which EAC cascade pathogenesis can be mediated. No studies have evaluated *Escherichia coli* Nissle 1917 in the EAC cascade up to this point. The analysis of *Escherichia coli* Nissle 1917 in the EAC cascade is subsequently a novel field that would benefit from analysis. 

## Figures and Tables

**Figure 1 biology-10-00697-f001:**
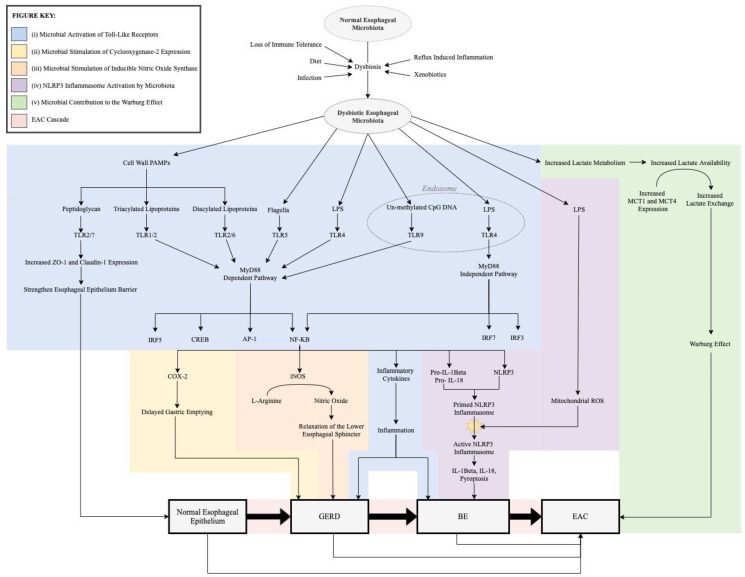
Molecular signaling in response to dysbiosis in EAC pathogenesis. The predominantly Gram-positive esophageal microbiome in the normal esophagus transforms through dysbiosis to a predominantly Gram-negative microbiome in the EAC cascade. The new microbiota can subsequently promote the pathogenesis of the EAC cascade through activation of toll-like receptors, stimulation of cyclooxygenase-2 expression, stimulation of inducible nitric oxide synthase, NLRP3 inflammasome activation, and contribution to the Warburg effect. PAMP = pathogen-associated molecular pattern; LPS = lipopolysaccharide; MCT = monocarboxylate transporter; TLR = toll-like receptor; ZO-1 = zonula occludens-1; MyD88 = myeloid differentiation primary response 88; IRF =interferon-regulator factor; CREB = cAMP-response element binding protein; AP-1 = activator protein 1; NF-kB = nuclear factor kappa B; COX-2 = cyclooxygenase-2; iNOS = inducible nitric oxide synthase; IL = interleukin; NLRP3 = nod-like receptor protein; ROS = reactive oxygen species; GERD = gastroesophageal reflux disease; BE = Barrett’s esophagus; EAC = esophageal adenocarcinoma.

**Table 3 biology-10-00697-t003:** Toll-like receptor characteristics [64,67].

Toll-Like Receptor		TLR 1 + TLR 2	TLR 2 + TLR 6	TLR 3	TLR 4	TLR 5	TLR 7	TLR 9
Cellular Expression	Monocytes	+	+		+	+	+	+
	Macrophages	+	+		+	+	+	+
	Dendritic cells	+		+	+	+	+	+
	NK cells			+				
	Mast cells		+		+			
	B cells	+	+	+			+	+
	T cells			+				+
	Esophageal epithelium	+	+	+	+	+	+	+
Membrane Expression		Plasma membrane	Plasma membrane	Endosome	Plasma membrane (Endosomal membrane)	Plasma membrane	Endosomal membrane	Endosomal membrane
PAMPs		Bacterial cell wall Triacyl Lipoproteins	Bacterial cell wall Diacyl lipoproteins Lipoteichoic Acid Zymosan	Viral dsRNA Viral ssRNA	Lipopolysaccharide Mannan Glycoinositolphospholipids Envelope Proteins	Flagellin	Viral ssRNA Imidazoquinolines	Unmethylated CpG DNAViral dsDNA
Downstream Signaling Pathway		MyD88 Dependent	MyD88 Dependent	MyD88 Independent	MyD88 Dependent (MyD88 Independent)	MyD88 Dependent	MyD88 Dependent	MyD88 Dependent
Downstream Signaling Factors	NFKB	+	+	+	+ (+)	+	+	+
	CREB	+	+		+	+		+
	AP-1	+	+		+	+		
	IRF	IRF5	IRF5	IRF3, IRF7	IRF5 IRF3, IRF7 (+)	IRF5	IRF7	IRF5, IRF7

Parentheses in association with TLR4 denotes the endosomal membrane expression of TLR4. TLR = toll-like receptor; NK cells = natural killer cells; PAMP = pathogen-associated molecular pattern; dsRNA = double stranded ribonucleic acid; ssRNA = single stranded ribonucleic acid; dsDNA = double stranded deoxyribonucleic acid; MyD88 = Myeloid differentiation primary response 88; NF-κB = Nuclear factor kappa B; CREB = cAMP-response element binding protein; AP-1 = Activator protein 1; IRF = Interferon-regulatory factor.

## Data Availability

Not applicable.

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
