# Peer review of "The Role of Microbiota in the Pathogenesis of Esophageal Adenocarcinoma"

_biology, 2021, doi:10.3390/biology10080697_

Round 1

Reviewer 1 Report

The authors have addressed points raised by the referees. However, there is still a problem in the manuscript with the page numbering from the diagram showing in figure 1. 

Author Response

Reviewer 1

Concern 1: The authors have addressed points raised by the referees. However, there is still a problem in the manuscript with the page numbering from the diagram showing in figure 1

  • Response: Thank you for raising this concern. We have corrected this error to the best of our abilities in the the newly submitted version of the manuscript. If submission to MDPI reverts the page numbers to the incorrect format, we will again request the Editor of Biology to resolve this issue.

Reviewer 2 Report

The authors have adequately addressed the concerns raised to their previous submission. The recent submission is much more comprehensive and well-rounded. I have no further comments.

Author Response

Reviewer 2

Thank you for your supportive comment.

Reviewer 3 Report

Manuscript ID: Biology-1271462 (previous submission Biology-1197960)

Title: The Role of Microbiota in the Pathogenesis of Esophageal Adenocarcinoma

Journal: Biology

Authors: Gillespie et al.

The above manuscript is very interesting. The new version of the manuscript is almost ready for publication. There are only minimal errors and deficiencies.

List of errors and shortcomings:

  1. 1. Introduction, previous comment 1, line 155.  The authors should consider the use of additional reference showing the role of gastrointestinal hormones and its effects on gastric secretion and motility, as well as the regulation of lower esophageal sphincter contraction (PMID: 25716961).
  2. Page 2, line 163. After “essentially” add an omitted word “empty”. Sorry for this.
  3. Page 2, line 163-164. Change the sentence “However, saliva…” to “However, saliva is produced and swallowed periodically also in inter-digestive period”.
  4. Page 2, lines 168. References cited by authors do not present the influence of saliva on healing effects in the oral cavity, stomach and duodenum. For this reason the authors should consider the use of additional references supporting their statement.
  5. Page 4-6, Table 1. The table header should be presented before the table, whereas, the table captions after the table. Therefore, the first sentence below the table should be presented as the table header before the table. The remaining elements can be presented as captions under the table.
  6. Page 5-8. Table 2. The same problem as in the case of table 1. See comment 5.
  7. Page ?, line 650. Change “pathogen associated” to „pathogen-associated”
  8. Page ?, line 652. Change “damage associated” to “damage-associated”.
  9. Still is observed the problem with text editing. The first 9 pages are numbered from 1 to 9. Figure 1 is presented on pages numbered as 0 of 30 and 1 of 30. Next three pages are again numbered as 0 of 30. Last 16 pages are numbered from 1 of 30 to 16 of 30. The order of line numbering is also not maintained. These errors should be corrected, but this problem should be solved by Biology editors during preparing the final form of the manuscript for publication.

Author Response

Concern 1: Introduction, previous comment 1, line 155.  The authors should consider the use of additional reference showing the role of gastrointestinal hormones and its effects on gastric secretion and motility, as well as the regulation of lower esophageal sphincter contraction (PMID: 25716961).

  • Response: Thank you for your comment and suggestions .We have included the suggested paper in the references.

Concern 2: Page 2, line 163. After “essentially” add an omitted word “empty”. Sorry for this.

  • Response: Thank you for bringing this edit to our attention. This paragraph has been reworded in the modified manuscript.

Concern 3: Page 2, line 163-164. Change the sentence “However, saliva…” to “However, saliva is produced and swallowed periodically also in inter-digestive period”.

  • Response: Thank you for your comment. This paragraph has been reworded in the revised manuscript.

Concern 4: Page 2, lines 168. References cited by authors do not present the influence of saliva on healing effects in the oral cavity, stomach and duodenum. For this reason the authors should consider the use of additional references supporting their statement.

  • Response: We appreciate your comment regarding these references. The references have been changed and the sentence has been altered to accurately reflect the referenced articles.

Concern 5: Page 4-6, Table 1. The table header should be presented before the table, whereas, the table captions after the table. Therefore, the first sentence below the table should be presented as the table header before the table. The remaining elements can be presented as captions under the table.

  • Response: Thank you for suggesting this edit. All table headers have been moved to precede the table. Captions are included below the corresponding table.

Concern 6: Page 5-8. Table 2. The same problem as in the case of table 1. See comment 5.

  • Response: Thank you for suggesting this edit. All table headers have been moved to precede the table. Captions are included below the corresponding table.

Concern 7: Page ?, line 650. Change “pathogen associated” to „pathogen-associated”

  • Response: Thank you for bringing this to our attention. This edit has been made throughout the article including the abbreviations section.

Concern 8: Page ?, line 652. Change “damage associated” to “damage-associated”.

  • Response: Thank you for bringing this to our attention. This edit has been made throughout the article including the abbreviations section.

Concern 9: Still is observed the problem with text editing. The first 9 pages are numbered from 1 to 9. Figure 1 is presented on pages numbered as 0 of 30 and 1 of 30. Next three pages are again numbered as 0 of 30. Last 16 pages are numbered from 1 of 30 to 16 of 30. The order of line numbering is also not maintained. These errors should be corrected, but this problem should be solved by Biology editors during preparing the final form of the manuscript for publication.

  • Response: Thank you for bringing this to our attention. We have corrected this error to the best of our abilities in the the newly submitted version of the manuscript. If submission to MDPI reverts the page numbers to the incorrect format, we will again request the Editor of Biology to resolve this issue.

This manuscript is a resubmission of an earlier submission. The following is a list of the peer review reports and author responses from that submission.

Round 1

Reviewer 1 Report

Manuscript ID: biology-1197960

Title: The Role of Microbiota in the Pathogenesis of Esophageal Adenocarcinoma

Journal: Biology

Authors: Gillespie et al.

The above manuscript is very interesting and presents the role of esophageal microbiota in esophageal pathology. The manuscript is well written. However, it must be noted that the manuscript contains some errors and shortcomings that should be corrected prior to publication.

List of errors and shortcomings:

  1. Section 1. Introduction. The authors should write a few more sentences on pathophysiology of GERD, the relationship between gastroesophageal reflux and GERD, as well as mechanism preventing GERD development (Zachariah et al, PMID: 32146942). They also should write about gastrointestinal hormones that affect gastric acid secretion and contraction of lower esophageal sphincter.
  2. Section 2. The authors should write a few words about the physiological role of saliva and its anti-inflammatory and anti-infective properties. The role of esophagus is to transfer food and fluids from the throat to the stomach. During the digestive period, the contact of the esophageal mucosa with ingested fluids, food and is short-term. The esophagus, apart from swallowing time, is essentially in physiological condition. However, saliva is also produced and swallowed periodically during the inter-digestive. Therefore, saliva plays an essential role in maintaining the integrity of the oral mucosa, as well as other parts of the digestive tract. Reducing the production of saliva or removal of the salivary glands leads to tooth loss and inflammation of the mucosa (Samnieng et al., PMID: 22482262), as well as reduced regeneration capacity within the oral cavity, esophagus (Rourk et al., PMID: 8304310), stomach and duodenum.
  3. Manuscript text editing problem. The first 4 pages are numbered from 1 to 4. Table 2 and Figure 1 are presented on the next 3 pages again numbered from 1 to 3. Next two pages are without page number (or numbered as page 1). Next 11 pages are again numbered from 1 to 11. This error should probably be fixed by editors.
  4. Page 3 line 120, Table 1, page 4 line 146, page 8 (page without page and lines numbers), and next pages. The authors presenting the names of the authors of the article in the text should always use a dot after “et al”. Moreover, the form “Peiet al.’s observation” (page 3 line 120) is strange and should be changed.
  5. Page 4, line 141. Replace “shoed” by “showed”.
  6. Page 4, line 154. “eac” should be replaced by “EAC”.
  7. Page 3? (6). Figure 1 legend. The sentence “The predominantly gram-positive esophageal microbiome present in the normal esophagus transforms through dysbiosis to a predominantly gram-positive microbiome in the EAC cascade” should be replaced by “The predominantly gram-positive esophageal microbiome present in the normal esophagus transforms through dysbiosis to a predominantly gram-negative microbiome in the EAC cascade”.
  8. Page 4 from the last 11 pages, line 9. The authors should replace the current form by “TNF-α”.
  9. Page 5 from the last 11 pages, the last paragraph. The authors should correct the sentence “Inflammasomes are expressed epithelial and immune cells”.
  10. The authors should write about some limitations of esophageal microbiome analysis performed by bacterial DNA extraction followed by amplicon sequencing targeting hypervariable region of the 16S rRNA gene. Extracted DNA mainly comes from live bacteria, but this method also isolates DNA from dead bacteria, as well as from bacterial residues. Thus, isolation of DNA specific for any bacterial strain may but not have to indicate the presence of live bacteria of that strain in the esophagus. This problem should be presented and discussed in the Discussion.
  11. In patients with GERD and BE, taking a small amount of cold reduces the feeling of heartburn. EAC leads to dysphagia, which causes patients to switch to a liquid, largely dairy-based diet. On the other hand, diet strongly modulates the gut microbiota composition (Wu et al., PMID: 21885731). These data indicate that an increase in abundance of lactic acid producing bacteria in the esophageal adenocarcinoma patients may be a symptom but not a mechanism of carcinogenesis in the esophagus. This problem should be presented in Discussion.
  12. In the discussion, the authors should write some sentences about the role of Helicobacter pylori in the human pathology, including the effects of Helicobacter pylori eradication of the incidence of GERD (Zhao et al., PMID: 32396919). Moreover, the authors should write that Helicobacter pylori, apart the stomach affects other organs, including the pancreas (Warzecha et al., PMID: 12138227).
  13. The authors’ results suggest that lactic acid producing bacteria may participate in the development of esophageal adenocarcinoma. Therefore, it is recommended that the authors present the other side of the problem and write a few words about potential protective and therapeutic role of probiotics in the digestive system, also taking into account the role of Escherichia coli Nissle 1917 in the protection and healing of the colon.

Author Response

Section 1. Introduction

Concern1:The authors should write a few more sentences on pathophysiology of GERD, the relationship between gastroesophageal reflux and GERD, as well as mechanism preventing GERD development (Zachariah et al, PMID: 32146942). They also should write about gastrointestinal hormones that affect gastric acid secretion and contraction of lower esophageal sphincter.

Response: Thank you for your comments and suggestions. We have modified the text and included more sentences regarding the relationship between gastroesophageal reflux and GERD, as well as mechanism preventing GERD development as well as the gastric hormones.

Concern 2: The authors should write a few words about the physiological role of saliva and its anti-inflammatory and anti-infective properties.

Response: Thank you for your suggestion. We have modified the manuscript accordingly and the text have been included (line 63-72) in the revised manuscript.

Concern 3: Manuscript text editing problem.

The first 4 pages are numbered from 1 to 4. Table 2 and Figure 1 are presented on the next 3 pages again numbered from 1 to 3. Next two pages are without page number (or numbered as page 1). Next 11 pages are again numbered from 1 to 11. This error should probably be fixed by editors.

Page 3 line 120, Table 1, page 4 line 146, page 8 (page without page and lines numbers), and next pages.

Response: Thank you for your comment. We have revised the manuscript provided by the journal and we hope the page number have been taken care of.

Concern 4: The authors presenting the names of the authors of the article in the text should always use a dot after “et al”. Moreover, the form “Peiet al.’s observation” (page 3 line 120) is strange and should be changed.

Response: Thank you for your comment. We have modified the manuscript accordingly.

Concern 5: Page 4, line 141. Replace “shoed” by “showed”.

Response: Thank you for your comment. We have replaced “shoed” by “showed” in the revised manuscript.

Concern 5:Page 4, line 154. “eac” should be replaced by “EAC”.

Response: Thank you for your comment. We have replaced “eac” by “EAC” in the revised manuscript.

Concern 6: Page 3? (6). Figure 1 legend. The sentence “The predominantly gram-positive esophageal microbiome present in the normal esophagus transforms through dysbiosis to a predominantly gram-positive microbiome in the EAC cascade” should be replaced by “The predominantly gram-positive esophageal microbiome present in the normal esophagus transforms through dysbiosis to a predominantly gram-negative microbiome in the EAC cascade”.

Response:  Thank you for your suggestion. We have reformatted the text as advised.

Concern7: Page 4 from the last 11 pages, line 9. The authors should replace the current form by “TNF-α”.

Response: Thank you for your comment. We have modified in the revised manuscript.

Comment 8: Page 5 from the last 11 pages, the last paragraph. The authors should correct the sentence “Inflammasomes are expressed epithelial and immune cells”.

Response: Thank you for your comment. We have modified the sentence.

Concern 9: The authors should write about some limitations of esophageal microbiome analysis performed by bacterial DNA extraction followed by amplicon sequencing targeting hypervariable region of the 16S rRNA gene. Extracted DNA mainly comes from live bacteria, but this method also isolates DNA from dead bacteria, as well as from bacterial residues. Thus, isolation of DNA specific for any bacterial strain may but not have to indicate the presence of live bacteria of that strain in the esophagus. This problem should be presented and discussed in the Discussion.

Response: Thank you for your comments. We have added the text in the Discussion section in the revised manuscript as suggested.

Concern 10: In patients with GERD and BE, taking a small amount of cold reduces the feeling of heartburn. EAC leads to dysphagia, which causes patients to switch to a liquid, largely dairy-based diet. On the other hand, diet strongly modulates the gut microbiota composition (Wu et al., PMID: 21885731). These data indicate that an increase in abundance of lactic acid producing bacteria in the esophageal adenocarcinoma patients may be a symptom but not a mechanism of carcinogenesis in the esophagus. This problem should be presented in Discussion.

Response: Thank you for your comments. We have added the text in the Discussion section in the revised manuscript as suggested.

Concern 11:In the discussion, the authors should write some sentences about the role of Helicobacter pylori in the human pathology, including the effects of Helicobacter pylori eradication of the incidence of GERD (Zhao et al., PMID: 32396919). Moreover, the authors should write that Helicobacter pylori, apart the stomach affects other organs, including the pancreas (Warzecha et al., PMID: 12138227).

Response: Thank you for your comments. We have added the text briefly in the Discussion section in the revised manuscript as suggested. We limits the discussion as detailed discussion will increase the length of the article and the our aim was to discuss changes in esophagus.

Concern 12: The authors’ results suggest that lactic acid producing bacteria may participate in the development of esophageal adenocarcinoma. Therefore, it is recommended that the authors present the other side of the problem and write a few words about potential protective and therapeutic role of probiotics in the digestive system, also taking into account the role of Escherichia coli Nissle 1917 in the protection and healing of the colon.

Response: Thank you for your comments. We have added the text briefly in the Discussion section in the revised manuscript as suggested.

Reviewer 2 Report

Gillespie et al., present a well-researched and well-articulated review on the role of microbiota in esophageal cancer pathogenesis. I think the authors have done a great job in succinctly summarizing several different aspects of the disease process. Each section flows smoothly and the summary tables and figures are quite informative. I only have a few minor suggestions. 

Line # 70-72 need more references. 

Line # 97-105 needs reference. 

The authors have shown a table (Table 1) with partial data on % distribution of several different organisms found in different studies. I can understand the lack of availability of relevant data in terms of getting absolute estimates for each of those organisms/studies. However, in my opinion, maybe it is helpful to expand a bit more specifically highlighting the quantitative nature of the dysbiotic shift that has been associated with EAC.    

Line 140, the word “showed” has a typo. 

Typo in the “findings” column for Yang et al., report. “Predominantly” is mis-spelt. 

The first line in the description for Fig. 1 (in the figure legend) ....I think the authors meant to say “.... transformed to gram-negative.....". Kindly revisit. 

Lack of page # line # makes it difficult to properly comment and review the manuscript. 

What about mice models of dysbiosis induced TLR2 abnormalities and changes in epithelium (apart from ref. 43, Zaidi et al.,2016?). Kindly re-visit. 

There is some discontinuity in page # and line #, but the part describing “proteobacteria.........PAMP for TLR5” needs a reference. 

NLRP3 inflammasome section could use some relevant primary references. Kindly re-visit. 

Maybe it is beneficial to add a note about H. pylori and its contentious association with EAC initiation.  Same could be said about E. coli.

Author Response

Concern 1:The authors have shown a table (Table 1) with partial data on % distribution of several different organisms found in different studies. I can understand the lack of availability of relevant data in terms of getting absolute estimates for each of those organisms/studies. However, in my opinion, maybe it is helpful to expand a bit more specifically highlighting the quantitative nature of the dysbiotic shift that has been associated with EAC.   

Response: Thank you for your suggestions. We agree that a more quantitative measure of the dysbiotic shift would be helpful, but the information available in the literature is very limited and remained unable to find quantitative method that could be used across all studies included in this section of the manuscript. We agree that this is a limitation.

Concern 2: Line 140, the word “showed” has a typo.

Response: Thank you for your comment. We have replaced “shoed” by “showed” in the revised manuscript.

Concern 3: Typo in the “findings” column for Yang et al., report. “Predominantly” is mis-spelt.

Response: Thank you for your comment. We have corrected it in the revised manuscript.

Concern 4: The first line in the description for Fig. 1 (in the figure legend) ....I think the authors meant to say “.... transformed to gram-negative.....". Kindly revisit.

Response: Thank you for your comment. We have modified the sentence in the revised manuscript.

Concern 5: Lack of page # line # makes it difficult to properly comment and review the manuscript.

Response: Thank you for your comment. We have included page number and line number in the manuscript.

Concern 6: Maybe it is beneficial to add a note about H. pylori and its contentious association with EAC initiation.  Same could be said about E. coli.

Response: Thank you for your comment. We have included the text rearding H. Pylori in the revised manuscript.

Concern 7: Line # 70-72 need more references.

Response: Thank you for your comment. We have added more reference to line 70-72

Concern 8: Line # 97-105 needs reference.

Response: Thank you for your suggestion. We have included the references.

Concern 9: There part describing “proteobacteria.........PAMP for TLR5” needs a reference.

Response: Thank you for your suggestion. We have included the reference.

Concern 10: NLRP3 inflammasome section could use some relevant primary references. Kindly re-visit.

Response: Thank you for your suggestion. We have included the references.

Concern 11: What about mice models of dysbiosis induced TLR2 abnormalities and changes in epithelium (apart from ref. 43, Zaidi et al.,2016?). Kindly re-visit.

Response: Thank you for your comment. We have modified the text and have included teat regarding mice model, dysbiosis, and TLR2 abnormalities.

Concern 12: TNM

Response: Thank you for your comment. We have defined TNM as "tumor, nodes, and metastasis".

Author Response

Concern 1: Lines 88-96. I would suggest rewording of this entire paragraph in order to eliminate apparent ambiguities. How can reconstitution reverse the pathological state if the underlying cause was a change in host immune tolerance? If dysbiosis is defined by alterations in the relative distribution of microflora, it seems obvious that altering the microflora affects dysbiosis (Lines 94-96, “This influence is demonstrated by the observation that reconstitution of the abolished commensal bacteria or their metabolites is capable of reversing the pathologic state of dysbiosis”). It is a circular statement. In context of the preceding sentence, it seems that the author’s meant this statement to address “..microbial functions that are vital to the host.” If so, then the specific microbial functions, or an example thereof, should be made explicit.

Response: Thank you for your suggestion. We have modified the text in the revised manuscript and have removed the sentence “This influence is demonstrated by the observation that reconstitution of the abolished commensal bacteria or their metabolites is capable of reversing the pathologic state of dysbiosis”.

Concern 2: Table 1. Row headings should be contained in the leftmost column for better readability and consistency with other tables. The results from Peter et al, 2020 seem to warrant inclusion in this table (and discussion in the text). Proportionate data (% representation of major genera) is not included for some studies (e.g. Dong et al 2018) even though it is present in the cited study(Supplementary Table 2).Peter S, Pendergraft A, VanDerPol W, Wilcox CM, Kyanam Kabir Baig KR, Morrow C, Izard J, Mannon PJ. Mucosa-Associated Microbiota in Barrett's Esophagus, Dysplasia, and Esophageal Adenocarcinoma Differ Similarly Compared With Healthy Controls. Clin Transl Gastroenterol. 2020 Aug;11(8):e00199. doi: 10.14309/ctg.0000000000000199. PMID: 32955191; PMCID: PMC7473866.

Response: Thank you for your comments and suggestions. We have modified the Table as suggested and have included the references in the table.

Concern 3: Line 141 shoed (sp)

Response: Thank you for your comment. We have corrected shoed as showed.

Concern 4: Figure 1. Title should include referral to host, as the figure is obviously not describing “microbial signalling” per se.

Response: Thank you for your suggestion. We have changed the title in the revised manuscript.

Concern 5: From now on, as line numbers are not present, suggested edits are located by the page number of the submitted pdf document (not page number on the document, as these are repeated). Page 8 “This pro-tumorigenic role (of) TLRs has been demonstrated in….”

Response: Thank you for your suggestion. We have modified the text in the revised manuscript.

Concern 6: Page 12 “ Previous studies have shown that TLR9 is capable of recognizing CpG DNA sequences present in bacteria and viruses as well as (and) parasitic hemozoin.” Note that hemozoin was later shown not to be a TLR9 ligand (Wu et al 2010), so the latter part of this statement is probably incorrect. Wu, X., Gowda, N. M., Kumar, S., & Gowda, D. C. (2010). Protein-DNA complex is the exclusive malaria parasite component that activates dendritic cells and triggers innate immune responses. Journal of immunology, 184(8), 4338–4348. https://doi.org/10.4049/jimmunol.0903824

Response: Thank you for your critical comment. We have corrected the sentence in the revised manuscript.

Concern 7: Page 13 TNF (alpha symbol has been altered)

Response: Thank you for your comment. We have modified it.

Concern 8: Page 13 TNM abbreviation introduced without definition

Response: Thank you for your comment. We have defined TNM as "tumor, nodes, metastasis".

Concern 9: Page 14. “Inflammasomes are expressed (in) epithelial and immune cells”.

Response: Thank you for your comment. We have modified the sentence as “Inflammasomes are expressed by epithelial and immune cells”.

Concern 10: Page 15 “Frist” (sp)

Response: Thank you for your comment. We have modified the text.

Concern 11: References. #15 contains incorrect DOI, #29 contains error (“&amp”), #63 contains incorrect journal abbreviation (correct in #74), #85 title and journal blended into single entry. Given the number of citation errors just from scanning the reference list, I suggest a thorough check.

Response: Thank you for your comment. We have modified the references in the revised manuscript. We have included the weblink for the articles where DOI is not available.